# Silica Hydrogels as Platform for Delivery of Hyaluronic Acid

**DOI:** 10.3390/pharmaceutics15010077

**Published:** 2022-12-26

**Authors:** Elena Parfenyuk, Ekaterina Dolinina

**Affiliations:** G.A. Krestov Institute of Solution Chemistry, Russian Academy of Sciences, 153045 Ivanovo, Russia

**Keywords:** anti-inflammatory drug, biodegradation, colloid silica, drug delivery, hydrogel, mechanical properties

## Abstract

Hyaluronic acid (HA) is chondroprotective and anti-inflammatory drug used clinically for treatment of inflammatory disorders (arthritis, skin diseases, bowel diseases, etc.). In addition, HA is a crucial ingredient in the cosmetic products used to eliminate the unpleasant consequences of inflammatory skin diseases. The main disadvantages that limit its use are its low mechanical properties and its rapid biodegradation. In this paper, silica hydrogels are considered as a promising matrix for HA to improve its properties. The hybrid HA-silica hydrogels were synthesized by the sol–gel method. Morphology of the hydrogels was investigated by optical microscopy and scanning electron microscopy methods. Taking into account their potential applications for topical and injectable delivery, much attention was paid to investigation of deformation properties of the hydrogels under shear, compression, and tension. Their resistance to enzymatic degradation in vitro was estimated. Kinetics and mechanisms of HA release from the hybrid hydrogels in vitro were also studied. It was found that the indicated properties can be controlled by synthesis conditions, HA molecular weight, and its loading in the hydrogels. Silica hydrogels are a prospective platform for the development of new soft formulations and cosmetic compositions of HA with improved pharmacological and consumer properties.

## 1. Introduction

Hyaluronic acid (HA) hydrogels are widely used in clinical practice and proposed for development of new drug formulations. Numerous studies have shown the enormous potential of hydrogels for the treatment of various diseases. For example, HA-based hydrogels are considered to be promising drug delivery systems for active targeting of malignant tumors [1,2], for intraocular delivery of antibodies [3], and for delivery of antibacterial drugs [4,5]. Hydrogels are of particular importance for the development of drug delivery systems for the treatment of inflammatory disorders, such as arthritis, skin diseases (atopic dermatitis, psoriasis, eczema, etc.), bowel diseases, etc. [2,4,6,7,8,9]. In addition, it is known that the acid itself has chondroprotective and anti-inflammatory effects. Due to its viscoelasticity, HA is a lubricant, which increases the viscosity of the synovial fluid and modulates inflammatory response [2,7,10,11,12]. Therefore, exogenous HA administered as an intra-articular injection may relieve the pain caused by inflammatory arthritis and slow progression of the disease. It was found that the inflammatory effect of HA depends on its molecular weight. High molecular weight HA has anti-inflammatory or immunosuppressive activity, whereas low molecular weight HA can induce pro-inflammatory or immunostimulatory responses [13,14]. Due to crucial role of HA in inflammatory response, as well as in other biological processes (cell adhesion, migration, proliferation, differentiation, angiogenesis, and tissue regeneration), HA hydrogels are used for development of new biomedical hydrogels for skin wound healing [13,14,15,16].

The special role of HA hydrogels in cosmetology should be noted. Owing to the great potential of HA in regulating inflammatory response, skin repairment, tissue regeneration, and immunomodulation, HA hydrogels have become the most crucial ingredients in cosmetic as well as nutricosmetic products (fillers, masks, gels, etc.) [17,18]. Inflammatory skin disorders can cause dryness, redness, and other unpleasant consequences [8]. In cosmetics, HA is widely used as powerful moisturizing agent due to its remarkable ability to hold/trap water [19,20]. Therefore, cosmetic HA-based formulations including hydrogels can help in the treatment of inflammatory skin diseases [8]. The effect of HA on the skin depends on the ability of HA to penetrate into the dermal layers. It is known that high molecular weight HA is unable to penetrate into the stratum corneum and mainly forms a protective film on the skin surface. In contrast, low molecular weight HA is able to engage in transdermal penetration [21,22]. Improvement of HA penetration into the stratum corneum and deeper dermal layers can be achieved by development of novel HA formulations, for example, emulsion [23], nanoparticulate poly-ion complexes of HA with poly-l-lysine [24], etc.

The treatment of inflammatory diseases using hydrogels often requires topical drug delivery or injections. For efficient application, HA hydrogels should have certain mechanical properties. Additionally, the resistance to degradation in biological media is very important property of the hydrogels intended for biomedical applications. However, it is known that although HA forms interconnected molecular networks in solution and the solutions exhibit viscoelastic properties [25,26,27], the network is characterized by short-lived structural integrity [28] and thus by weak mechanical stability. Exogenous HA undergoes rapid degradation in vivo, which significantly reduces the period of its functioning. The degradation may be induced by enzymes, pH, free radical stress, etc. [29].

The mechanical stability and degradation rate of HA can be improved by its crosslinking. Various strategies for physical and chemical crosslinking of HA have been developed [2,4,5,16]. In most of these strategies, crosslinking of HA chains occurs with participation of organic agents. As a result, a purely organic hydrogel is formed. In the last decade, numerous studies have shown the prospects of biomedical applications of hybrid hydrogels of HA prepared using colloid silica. Colloid silica is known to be a biocompatible, low-toxic, slowly biodegradable material in biological media. Silica is not subject to enzymatic degradation and is resistant to microbial attacks [30,31]. It was shown that the addition of pre-synthesized silica nanoparticles as inorganic filler to HA hydrogels led to the enhancement of the mechanical properties of the hydrogels [32]. Piantanida et al. developed nanocomposite HA-based hydrogels with thiol modified mesoporous silica particles covalently attached to the polymeric network. The obtained hydrogels formed in situ and had mechanical properties able to stimulate the healing process of esophageal fistulas in animals [33]. Sánchez-Téllez et al. used silica as a crosslinking agent to prepare hybrid HA-silica hydrogels. HA was chemically modified with an aminoalkoxysilane and crosslinked using tetraethoxysilane (TEOS) and polydimethylsiloxane via sol–gel reactions [34]. The hybrid HA–silica hydrogels with pre-crosslinked HA (1,4-butanediol diglycidyl ether (BDDE) as the chemical crosslinker) were synthesized using 3-glycidyloxypropyl-trimethoxysilane (GPTMS) for silanization [35]. It was shown that the sol–gel stabilized HA hydrogel exhibited superior mechanical properties and biochemical stability as well as excellent biocompatibility without triggering any negative biological effects. The nanohybrid hydrogels of HA crosslinked by BDDE were prepared using tetramethoxysilane (TMOS) as a silica precursor via the in situ sol–gel process. The hybrid hydrogels exhibited significantly enhanced mechanical properties compared to the polymer sample, with great degradation resistance and bioactivity. The authors of the study note that these characteristics can be modulated by varying the silica content [36]. It should be noted that in most works devoted to synthesis and investigation of the hybrid HA–silica hydrogels, the acid was preliminarily crosslinked using organic crosslinkers. Flegeau et al. developed in situ forming injectable HA–silica hydrogels without additional crosslinking of the polymer [37]. HA was chemically modified with an aminoalkoxysilane, and homogeneous hydrogels were obtained upon simple pH adjustment of the silanized HA solution (pH 12.3) to physiological pH. It was shown that the mechanical properties and degradability of the hydrogels depend on the degree of HA silanization and molecular weight of the acid.

In this work, silica hydrogels are considered to be a promising platform for development of new HA soft formulations and cosmetic compositions. In our previous work, using small drug molecules as an example, we have shown that silica hydrogels are a promising platform for the development of delivery systems for topical and injectable administration [38,39]. In the present work, the large and more complex molecules of hyaluronic acid were embedded in silica hydrogels. It should be noted that generally, in the hybrid HA-silica hydrogels reported in the literature, the polymer phase acts as a hydrogel matrix in which the inorganic component is embedded. The hydrogels prepared in this way are inorganic/organic hybrid materials. The hybrid hydrogels synthesized in this work are considered to be organic/inorganic, in which silica hydrogel is the main matrix. The hydrogel materials were prepared without additional HA modification or crosslinking using TEOS as silica precursor. The hybrid hydrogels were synthesized using different catalyst concentrations of silica sol formation (HCl) and different HA loading in the hydrogels. HA with different average molecular weights was investigated (50–100 kDa and 1000–1500 kDa). Although, as indicated above, only high molecular weight HA (>1000 kDa) has anti-inflammatory properties, low molecular weight HA was investigated to identify the effects of its molecular weight on the properties of HA–silica hybrid hydrogels. All the synthesized hydrogels had a pH of 6.4–6.8.

Taking into account the potential applications of the hybrid hydrogels as soft drug formulations and cosmetic products, in this work much attention was paid to the study of deformation properties of the hydrogels under shear, compression, and tension. These properties determine the convenience and safety of administration (topical, injection) of hydrogel drug formulations and, as a result, their biological efficacy [40,41]. Additionally, the deformation properties greatly influence the manufacturing process of the pharmaceutical products because the properties allow optimization of various stages of the process: mixing, pumping, transfer, packing, etc.

Another important task of this work was to study the resistance of the synthesized hybrid hydrogels to enzymatic degradation of HA. It is well known that HA undergoes hydrolytic degradation by specific enzymes called hyaluronidases [29]. The enzymes cleave the glycosidic bonds of the polymeric backbone, resulting in the appearance of HA fragments with lower molecular weight. The decrease in polymer chain length is responsible for the reduction in viscosity observed in these systems. Therefore, rheology is an appropriate method to study the enzymatic degradation of polymers [42,43,44]. The rheological study of the synthesized HA–silica hydrogels has shown whether the hydrogel silica matrix is capable of slowing down the process of enzymatic degradation of HA in the hybrid hydrogels.

Kinetics of drug release is very important functional property of any drug delivery system. In the present work, the kinetics of HA release from the synthesized hybrid hydrogels was investigated in the media with pH 5.5 (32 °C) and 7.4 (37 °C), i.e., in the media simulating the skin surface and blood plasma.

In order to optimize the conditions of synthesis of the hybrid hydrogels, the effects of HA molecular weight, the catalyst concentration of silica sol formation and HA loading on the deformation properties, degradation stability of the synthesized hydrogels, and the kinetics of HA release from them were revealed.

## 2. Materials and Methods

### 2.1. Materials

Hyaluronic acid sodium salt (HA)(MW 50–100kDa and 1.0–1.5 MDa) was purchased from Glentham Life Science Ltd. (Corsham, UK). Tetraethoxysilane (TEOS) (high purity grade, ECOS, Moscow, Russia) was used as a silica precursor. Hydrochloric acid (HCl) (for analysis, 37%, Acros Organics, Geel, Belgium, CAS 7647-01-0) was used as a catalyst for silica sol formation. Acetic acid (99.7%+, Acros Organics, CAS 64-19-7) and sodium hydroxide (NaOH) (analytical grade, Chimmed, Moscow, Russia) were used to prepare buffer solutions. Sodium dihydrogen phosphate (NaH_2_PO_4_·2H_2_O) and disodium hydrogen phosphate (Na_2_HPO_4_·12 H_2_O) (analytical grade, Chimmed, Moscow, Russia) were used to prepare a buffer solution with pH 7.4. Hyaluronidase (205.5U/mg) was isolated from the testes of cattle (Microgen, Russia). Stains-all (97%, Acros Organics, CAS 7423-31-6, NJ, USA) was used for determination of the amount of released HA. Potassium bromide (KBr), 99+%, IR grade, Acros Organics, Geel, Belgium) was dried at 250 °C before use.

### 2.2. Synthesis of Hydrogel Materials

First, a weighted sample of HA was dissolved in 50 mL of a phosphate buffer solution (pH 7.4). The solution was kept for a day to swell the acid. Next, silica sol was prepared by mixing TEOS (5 mL), water (1 mL), and a catalyst (HCl solution with a concentration of 0.03 M or 0.125 M) (1 mL). The prepared sol was added dropwise to the buffer solution. After homogenization, the prepared sol was added dropwise to the HA solution with vigorous stirring for ~1.0–1.5 h. After gelation, the obtained hydrogel composites were aged for 10 days and stored in tightly closed containers.

The pure silica hydrogels were prepared using the procedure described above but without addition of HA.

The designation and synthesis conditions of the hydrogel materials are presented in Table 1.

### 2.3. Characterization

#### 2.3.1. Fourier–Transform Infrared (FTIR) Spectroscopy

The FTIR spectra were recorded using a VERTEX 80v (Broker Optik, Germany) spectrometer at room temperature. The spectra were recorded in the range of 4000 to 400 cm^−1^. The samples were examined as KBr disks.

#### 2.3.2. Morphology

To investigate the morphology of the hydrogels containing a large amount of liquid phase in their original form, optical microscopy images were obtained using an optical microscope XSP-104 equipped with a Micro Ocular PCE-ME 100 camera (Armed, Russia). In addition, the dried hydrogels were studied using a Quattro S scanning electron microscope (Thermo Fisher Scientific, Czech Republic). The hydrogels were dried at 1500 °C under vacuum.

#### 2.3.3. Rheological Study

Steady-state flow measurements were performed at room temperature using a Brookfield DV2T Viscosimeter (Brookfield, AMETEK, Inc. Middleborough, MA, USA). A hydrogel sample was placed in a cylindrical cell, and a spindle attached to the device was accurately immersed into the cell. The hydrogel was equilibrated for 10 min prior to testing. Viscosity and shear stress were measured in a shear rate range of 0.3–6.3 c^−1^.

The obtained flow curves were fitted with the modified Bingham (1) model [45,46] as well as the Bingham (2), the Casson (3), the Ostwald-de Waele or Power Law (4), and the Herschel-Bulkley (5) models [47]:(1)$$\tau =\tau_{0}+\eta_{B}\gamma +C\gamma^{2}$$
(2)$$\tau =\tau_{0}+\eta_{B}\gamma$$
(3)$$\tau^{0.5}=\tau_{0}^{0.5}+{\left(\eta_{c}\gamma \right)}^{0.5}$$
(4)$$\tau =K\gamma^{n}$$
(5)$$\tau =\tau_{0}+K\gamma^{n}$$
where γ is the shear rate, τ_0_ is the dynamic yield stress, and η_B_ and η_C_ are the Bingham and Casson plastic viscosities, respectively; C is the constant, K is the consistency index, and n is the flow behavior index.

The thixotropic properties were studied using the method of a hysteresis loop. In this experiment, the apparent viscosity vs shear rate dependences were recorded at the increasing (from 0 to 6.3 s^−1^) and then immediately decreasing (from 6.3 to 0 s^−1^) shear rate. The hysteresis loop area can be found through the integration method:(6)$$S=\int_{\gamma_{1}}^{\gamma_{2}}(Y_{up}-Y_{down})d\gamma$$
where γ_1_ and γ_2_ are the initial and final shear rates, and Y_up_ and Y_down_ are the functional relationship of the up and down curves [48,49]. In the present work, the degree of thixotropy was estimated as the thixotropy index T [50,51], which is defined as
(7)$$T=\frac{S_{up}-S_{down}}{S_{up}}$$

The areas were calculated using the trapezoidal method [52,53].

#### 2.3.4. Compression and Tension Tests

Uniaxial compression and tension tests were conducted at room temperature using a test machine [54]. Samples of height (H = 5 mm) and diameter (D = 20 mm) were used. The tests were performed at a constant cross-head speed of 0.021 mm·s^−1^. The compression strain (ε_c_) and the tensile strain (ε_t_) were determined as ε_c_ = 1 − λ and ε_t_ = λ − 1, respectively, where λ = l/l_0_ (l_0_ and l are the heights of sample before and after deformation). The compression stress (σ_c_) and the tensile stress (σ_t_) were calculated as σ_c_ = F_c_/A_0_ and σ_t_ = F_t_/A_0_, respectively, where F_c_ and F_t_ are the load forces at compression and tension and A_0_ is the surface area. The compressive and tensile Young’s modulus values were calculated from the slope of the initial linear part of the stress–strain curves. The ultimate compressive and tensile strength values were determined as the maximum stress on corresponding stress–strain curves.

#### 2.3.5. In Vitro Enzymatic Degradation Study

The enzymatic degradability of the synthesized hydrogels was estimated by the rheological method using a Brookfield DV2T Viscosimeter (Brookfield, AMETEK, Inc. Middleborough, MA, USA). For this purpose, hyaluronidase was dissolved in isotonic solution (NaCl, 0.9%) to prepare 2158 U/mL. Then an appropriate amount of the solution was mixed with a hydrogel to obtain 22U/mL of the hydrogel. At pre-specified time points, a sample of the hydrogel was taken for rheological measurement. The apparent viscosities were determined at a constant shear rate of 0.5 s^−1^. The degree of viscosity loss was calculated by using following equation:(8)$$\eta \%=\frac{\eta_{0}-\eta_{t}}{\eta_{0}}\times 100$$
where percent η% is the percent viscosity lost after time t; η_t_ is the final viscosity at time t; and η_0_ is the initial viscosity of the gel [43]. In order to take into account the contribution of the shear-thinning effect to the loss of apparent viscosity, the measurements of apparent viscosity under the same conditions for the samples without the enzyme addition were also carried out. Due to the low shear rate, the contribution of the thixotropy of the samples, as well as the absence of shear load between measurements at a certain time (the measurements were carried out for 8 min), were found to be negligible (1–2%).

#### 2.3.6. In Vitro Release Kinetics Measurements and Analysis of the Obtained Data

The kinetic release profiles of HA from the hydrogel composites were obtained by incubation of 0.5 g of the hydrogel in 100 mL of aqueous media simulating the pH of skin surface and blood plasma. At appropriate time intervals, 5 mL samples were withdrawn and replaced by fresh release medium. The withdrawn samples were centrifuged at 10,000 rpm for 10 min. The release was studied in the acetate buffer with pH 5.5 at a temperature of 32 °C (simulation of topical administration conditions) or in water with addition of NaOH with pH 7.4 at a temperature of 37 °C (simulation of release after injection) under stirring (100 rpm). Such release media were chosen because the amount of released HA was determined using Stains-All dye. The addition of some anions to the dye solution leads to its aggregation and, as a result, a change in the position of the absorption maximum of the dye. The effect is very prominent for citrate, sulphate, and phosphate ions, but relatively weak for nitrate and acetate ions [55,56]. Stains-All forms a complex with glycosaminoglycans whose optical properties are different from those of the free dye [57,58]. Stains-All (5 μg) was dissolved in a water (40 mL) + ethanol (10 mL) mixture. The dye solution was added to the withdrawn sample in a ratio of 1:2 (*v*/*v*). The amount of released HA was calculated from the absorbance at a wavelength of 640 nm measured spectrophotometrically (UV/VIS SF-2000 spectrophotometer, LOMO, St. Petersburg, Russia) using the calibration curves obtained at a given pH. The release profiles were obtained by plotting the cumulative percentage of released drug as a function of release time.

The release profiles were fitted with various kinetic models: the zero-order model (9), the first order model (10), the Hixon–Crowell model (11), and the Korsmeyer–Peppas model (12) [59]:(9)$$Q_{t}=Q_{0}+k_{0}t$$
(10)$$Q_{t}=Q_{0}\cdot e^{-k_{1}t}$$
(11)$$Q_{0}^{1/3}-Q_{t}^{1/3}=k_{H-C}t$$
(12)$$\frac{M_{t}}{M_{\infty }}=kt^{n_{K-P}};\frac{M_{t}}{M_{\infty }}\leq 0.6$$
where k_0_, k_1_, k_H_-_C_, and k, are the zero order, the first order, the Hixon–Crowell, and the Korsmeyer–Peppas constants, respectively; Q_o_ and Q_t_ are the initial amount of drug in solution and the cumulative amount of drug released at time t, respectively; M_t_ and M_∞_ are the cumulative amount of released drug at time t and overall released amount, respectively; n_K–P_ is the diffusion exponent (related to the drug release mechanism) in the Korsmeyer–Peppas model.

#### 2.3.7. Statistical Analysis

The measurements of the deformation characteristics were repeated five times. The release measurements were carried out in triplicate. The obtained results are presented as an average of the indicated measurements ± standard deviation. The description of the experimental rheological data by various mathematic models was carried out using Rheocalc software RHEOCALC T 2.1.51 (Brookfield). The predicted rheological and release data for each model were evaluated using the coefficient of correlation (R^2^). The best fitting of the models was based on the highest R^2^.

## 3. Results

### 3.1. Synthesis of Hydrogels and Their Visual Observations

The hydrogels were synthesized by the sol–gel method. The prepared acidic silica sol (~pH 3) was poured into HA solution of a given concentration. Taking into account the potential applications of the hydrogels, HA was dissolved in the buffer solution with pH 7.4 to neutralize the acidic silica sol and to obtain the final hydrogels with pH 6.4–6.8. The prepared hydrogels were smooth, uniform, and opalescent materials. During the aging process, some hydrogels released some amount of liquid phase due to shrinking, but after stirring or shaking, the released liquid was incorporated into the gel structure, and the hydrogels became uniform again. The pure silica hydrogels and the hydrogels prepared using the high molecular weight HA did not flow down when turned over, while other hydrogels flowed slowly.

### 3.2. FTIR Study

Figure 1 shows the FTIR spectrum of solid HA, as well as the spectra of dried silica (HG1) and HA–silica (HG1(2%)l) hydrogels as examples.

It can be seen that the spectrum of the composite hydrogel exhibits characteristic bands of both the silica matrix and HA. The presence of HA is indicated by the band at 2928 cm^−1^ associated with the stretching vibrations of C–H in alkyl groups, the shoulder at 1715 cm-1 corresponding to the stretching C=O vibrations, and the band at 1624 cm^−1^ (Amide I) and the shoulder at 1567 cm^−1^ (Amide II) [60] shifted due to noncovalent interactions of the polymer with the silica matrix. The bands at 1413 cm^−1^ and 1378 cm^−1^ can be associated with C–H bending vibrations or C–O stretching and O–H bending vibrations [60].

### 3.3. Morphology

The morphology of the synthesized hydrogel materials was investigated by optical microscopy. The method allows us to get an idea of the morphology of the hydrogels in their original form. As an example, the optical images of HG2(1%)h and HG2(1%)l hydrogels are presented in Figure 2.

Other samples had a similar morphology. The optical images show that the surface of the hydrogels is porous. The pores have a round and slit-like shape. The porosity is higher for HG2(1%)l.

Figure 3 shows the SEM images of dried pure and some composite hydrogels. The SEM images of the dried pure silica hydrogels prepared using HCl concentration 0.03 M and 0.125 M (HG1 and HG2) (Figure 3a,b) demonstrate that the hydrogels are particulate materials. The structure of HG1 consists of aggregates of irregular shape and size (Figure 3a), whereas HG2 exhibits a more spherical randomly ordered morphology (Figure 3b). The hybrid polymer–silica hydrogels also exhibit a particulate nature. However, the microstructure of the hybrid HA–silica materials differs significantly from the structure of pure silica inorganic hydrogels. Figure 3c–f shows that the hybrid materials are denser, and individual grains can be observed. The grains exhibit a rough surface formed by interconnected clusters of the spherical particles and are cut by the channels formed by interconnected pores. As can be seen from Figure 3f, the HG2(2%)h has a smoother, denser, less porous surface and a special layered morphology. Thus, the SEM investigation showed that the catalyst concentration and HA molecular weight had a significant impact on the morphology of the materials obtained.

### 3.4. Rheological Properties

As has been mentioned above, the deformation properties of hydrogels under shear are very important for their manufacturing process and biomedical applications. As an example, Figure 4 shows the dependencies of apparent viscosity (η_app_) on shear rate for some HA–silica hydrogels and pure silica hydrogels. The dependencies for other samples are similar to those shown in Figure 4.

The apparent viscosity decreases with an increase in the applied shear rate, i.e., the synthesized hydrogels exhibit a shear-thinning effect (pseudoplasticity). It is well known that aqueous solutions of HA are pseudoplastic fluids [61,62,63]. Their shear-thinning behavior is due to deformation and alignment of the polymer chains in the streamlines of flow, resulting in the viscosity decrease [61]. The silica hydrogels also exhibit pseudoplasticity, which is explained by breakdown of their three-dimensional network and aggregates under increasing shear rates and organization of the formed smaller particles in the flow direction [64]. Therefore, it was expected that the composite HA–silica hydrogels exhibit pseudoplasticity.

The description of the experimentally obtained flow curves (τ = f(γ)) by various models showed that the Bingham, the modified Bingham, and the Casson models exhibited poor description of the curves (R^2^ = 0.58–0.91). A good fit with the experimental data was observed for the Ostwald–de Waele model (R^2^ = 0.93–0.98). The best fit was observed when using the Herschel–Bulkley model (R^2^ = 0.96–0.99).

The description of the flow curves by the Herschel–Bulkley model made it possible to simultaneously determine the values of the most important rheological characteristics: yield stress (τ_0_), flow index (n), and consistency index (K). The yield stress determined by model fitting is dynamic yield stress [65,66,67]. For thixotropic material, dynamic yield stress is related to the structure of the material destroyed by shear forces and requires minimum stress to maintain or terminate the material flow [66,67]. The flow index n indicates the non-Newtonian or Newtonian character of the material (for non-Newtonian pseudoplastic and dilatant systems n < 1 and n > 1, respectively; for Newtonian systems n = 1) and is a quantitative characteristic of degree of deviation of the system from Newtonian behavior. The lower the value of n, the more shear-thinning the system is [47]. The consistency index K is related to the material’s viscosity [47].

Figure 5a shows the dynamic yield stress values that characterize the structure of the hydrogels subjected to shear deformation (for example, the structure of a hydrogel squeezed out of a syringe).

It is seen that the τ_0_ values for pure silica hydrogels synthesized by the indicated method are very low. The particulate silica hydrogels quickly lose their original 3D structure under the action of shear forces, which leads to the formation of individual particles and their aggregates. This is evidenced by very low values of flow behavior index n (Figure 5b), which indicate large shear-thinning effects of these hydrogels under the influence of deformation forces.

The incorporation of HA into the silica hydrogels leads to an increase in the dynamic yield stress (Figure 5a). This means that a higher shear stress is required to maintain flow of HA–silica hydrogels in comparison to pure silica hydrogels because after the flow stops, the hybrid hydrogels have a stronger structure. This effect is significantly higher for high molecular weight HA and increases with increasing HA loading and concentration of HCl used for preparation of the silica sol. The n values for HA–silica hydrogels indicate that the hydrogels, like pure silica hydrogels, show a shear-thinning effect (pseudoplasticity) (n < 1). However, the n values are significantly higher in comparison with pure silica hydrogels (Figure 5b). This indicates a lower shear-thinning effect of the hybrid hydrogels. These effects can be associated with the HA–silica interactions in the hybrid hydrogels. It should be noted that the n values decrease with increasing polymer concentration and its molecular weight in the hybrid hydrogels, which show an increase in their non-Newtonian behavior. Furthermore, the flow behavior index can be considered as the rate of change of a structure with shear rate or shear stress: the higher the n, the less stable is the structure of the hydrogels under shear load [68,69,70].

The consistency index K is an indicator of the viscous nature of the studied systems. The calculated values of K for the synthesized hydrogels changed within the range of 2.04–12.76 Pa·s^n^. The values increased with the increasing molecular weight of HA and its loading.

Thixotropy is the most important rheological property of pharmaceutical and cosmetic products. The obtained results showed that all synthesized hydrogels are thixotropic. As an example, Figure 6 shows hysteresis loops of some synthesized hydrogels.

It is interesting that HA solutions do not exhibit thixotropic properties [23], while the silica hydrogels are thixotropic. Thus, the introduction of HA into silica hydrogel led to formation of thixotropic hydrogel materials.

The thixotropic properties of the synthesized hydrogels were estimated using the thixotropic index (T), which is an indicator of the extent of the thixotropic nature of the hydrogels. The higher the T value, the more time is required to restore the structure of the hydrogel, i.e., the more thixotropic the hydrogel is. The obtained data presented in Table 2 show that thixotropy of the hydrogels depends on HA molecular weight, its loading, and concentration of the catalyst of silica sol formation.

### 3.5. DeformatWion Properties under Compression and Tension

Along with the deformation properties under shear, the deformation properties of the synthesized hydrogels under compression and tension are essential characteristics for design of soft drug formulations and their applications. Typical experimental stress–strain curves obtained in uniaxial compression and tension tests for hydrogels are presented in Figure 7.

The obtained stress–strain curves allow for the calculation of the most important deformation characteristics. The compressive and tensile Young’s moduli are characteristics of the ability of a material to withstand changes in length when it is under lengthwise compression or tension and measure the compressive or tensile stiffness of the material. The ultimate tensile or compressive strength is the maximum stress that a material can withstand under a tensile force without breaking. The indicated characteristics of the synthesized hydrogels are presented in Table 3.

The obtained data show that the synthesized hydrogels are characterized by low tensile and compressive YM and ultimate strengths. This means that the hydrogels deform easily under the WWloads. This can be associated with large amounts of aqueous phase retained in the materials. The pure inorganic hydrogels show the highest compressive moduli and the lowest tensile moduli. Apparently, this is due to the strong matrix formed by the condensation of silica particles, as well as its high porosity. HA incorporation into the pure silica hydrogels leads to an increase in the tensile properties and a decrease in the compression properties of the hybrid hydrogels. The incorporation of HA results in partial replacement of strong covalent siloxane bonds (Si–O–Si) by less strong noncovalent HA–silica matrix interactions. The more acid is loaded into the silica hydrogel, the less strong its structure becomes. The structure of the formed hybrid hydrogels becomes less strong and more flexible. As a result, their mechanical properties under compression deteriorate, but their elasticity in tension increases.

However, the deformation properties of HA–silica hydrogels are significantly higher in comparison with pure HA. As can be seen from the data presented in Table 3, in general, the resistance to compressive forces of the hybrid hydrogels decreases with increasing HA molecular weight and its loading.

The tensile YM decreases with increasing HA loading. The ultimate tensile strength values are close for all HA–silica hydrogels, i.e., they begin to break down at the same tensile stress. The ultimate compressive strength, like the compressive YM, decreases with increasing HA molecular weight and its loading.

### 3.6. Enzymatic Degradation Study

HA is easily degraded by enzymes in vivo, which significantly reduces its functioning time. HA has a half-life of 3 to 5 min in the blood, less than a day in the skin, and 1 to 3 weeks in cartilage [71]. Degradation of HA leads to a decrease in the molecular weight and, consequently, to a decrease in viscosity. Therefore, the degradation of the synthesized hybrid hydrogels was analyzed by measuring their apparent viscosity at a constant shear rate. The degradation rate was estimated as the degree of decrease in the viscosity of the hydrogel with an increase in the time of hydrogel incubation in the presence of an enzyme. Hyaluronidase as a representative of enzymes present in the body’s fluids was chosen to study the degradation of HA in the synthesized hydrogels. Figure 8 shows the degree of apparent viscosity loss (%) for the hybrid hydrogels containing 2% of HA.

As can be seen from Figure 8, the high molecular weight HA exhibits a sharper decrease in apparent viscosity compared to the low molecular weight HA. It is also seen that the rate of viscosity loss for HG1(2%)l and HG2(2%)l is higher than for pure low molecular weight HA (Figure 8a), i.e., the acid in the hydrogels degrades faster. The effect is especially evident for HG2(2%)l. In contrast, the rate of apparent viscosity loss of the hybrid hydrogels containing the high molecular weight HA is lower than for pure acid, especially during the first 20 h (Figure 8b). This means that silica matrix is able to increase the resistance of the high molecular weight HA in the hybrid hydrogels to enzymatic degradation. The obtained results show that the enzymatic degradation of the prepared hydrogels depends on their synthesis conditions and the molecular weight of HA.

### 3.7. In Vitro Release Kinetics of HA from Synthesized Hybrid Hydrogels

Figure 9 shows the experimental dissolution profiles of HA and release profiles of HA from the synthesized hybrid hydrogels into the media with pH 5.5 (32 °C) and 7.4 (37 °C), simulating the skin surface and blood stream. It is seen that the hydrogels demonstrate delayed release of HA. In general, the release curves exhibit high burst effects (12–70%)., which were observed for high molecular weight HA and in the release medium with pH 7.4. Such high burst effects can be associated with breakdown of the 3D silica network of the hydrogels and formation of silica particles and their aggregates and agglomerates, resulting in release of weakly-bonded HA.

The profiles after the burst effects were fitted with kinetic models (9)–(11). According to the correlation coefficient values (R^2^), the zero order and the Hixon–Crowell models are inapplicable to the description of the obtained release profiles (R^2^ = 0.45–0.80). The first order model showed the best fit with the experimental curves (R^2^ = 0.98–0.99). A good fit was observed for the Korsmeyer–Peppas model (R^2^ = 0.95–0.98). The kinetic parameters of HA release calculated using the latter models are summarized in Table 4. The values of diffusion exponent n in the Korsmeyer–Peppas model indicate that the release process is controlled by pseudo-Fickian or retarded diffusion [72,73]. Apparently, after the burst effects, HA is released from the formed agglomerates and aggregates. The bad fit of the experimental curves with the Hixon–Crowell model testifies about the absence of significant changes in their surface area and diameter during the release, i.e., further breakdown of the agglomerates and aggregates does not occur. The pseudo-Fickian diffusion may be due to their polydispersity [74]. In addition, delayed diffusion may be associated with the large size of the polymer molecules and their interactions with the silica matrixes. It is unexpected that, in general, the rate constants of release and the maximum amounts of the released HA for 2 days are higher for the high molecular weight HA than those for the low molecular weight HA (Figure 9, Table 4). Perhaps this is connected with the weakening of HA–silica interactions due to the entanglement of the polymer chains and decreasing active binding sites of HA. According to data from the literature, the effect of entanglement is the most pronounced for high molecular weight HA [75]. Another reason may be the higher amount of water retained by high molecular weight HA, which promotes faster “leaching” of HA from the silica matrix. It is likely that the low molecular weight HA interacts more strongly with the silica matrices and, therefore, shows lower k_1_ values. The decrease in the rate constants with increasing HA loading can be connected with stronger intermolecular interaction between the polymer molecules, which inhibits their diffusion.

## 4. Discussion

The purpose of this study was to evaluate the possibility of application of silica hydrogels as a basis for soft anti-inflammatory formulations of HA.

Silica hydrogel is a colloid system consisting of a 3D framework formed from silica particles interacting with each other, the voids (pores) of which are filled with aqueous medium [76,77]. Consisting of colloid silica particles and retaining a lot of water, silica hydrogels are biocompatible, have low toxicity, and are stable to enzymatic, microbial, and bacterial attacks. Due their high porosity, the hydrogels are able to load large amounts of drugs and can be used for drug delivery. In addition, synthesis of silica hydrogels is inexpensive. Thus, like solid silica particles, silica hydrogels are promising materials for development of drug delivery systems, including anti-inflammatory drugs.

The treatment of inflammatory diseases often requires topical or injectable administration of the drugs. Therefore, anti-inflammatory soft drug formulations such as ointments, gels, creams, emulsions, etc. are widely used in clinical practice. The deformation properties of the formulations under shear, compression, and tension are very important from the point of view of convenience and safety of their administration, as is optimization of different steps of their manufacture process (mixing, pumping, packing, etc.).

The rheological study of the synthesized pure silica and hybrid hydrogels showed that they exhibited low dynamic yield stress values, which increased with increasing HA molecular weight and its loading. For the convenience of topical or injectable administration, dynamic stress should not be high so as not to apply a lot of effort to maintain the flow of a hydrogel from a syringe or when spreading to the skin. Additionally, the hydrogels exhibited a shear-thinning effect (pseudoplasticity) and thixotropy. The highest shear-thinning effect was observed for pure silica hydrogels (Figure 5). The 3D structure of the hydrogels is rapidly destroyed under action of shear forces, leading to the formation of suspension of silica particles and their aggregates [64]. Therefore, the restoration of the structure with a decrease in shear load occurs slowly, i.e., pure silica hydrogels have high thixotropic properties (Table 2). The introduction of HA in the silica hydrogels led to a decrease in the shear-thinning effect and, in general, to a decrease in their thixotropy. This can be due to the polymer–silica interactions in the hybrid hydrogels, which increase with increasing molecular weight of HA and its loading in the hybrid hydrogels. It should be noted that HA itself is not thixotropic. The pseudoplasticity and thixotropy are desirable for soft drug formulations. Due to these properties, HA–silica hydrogels are more stable than pure HA to be retained in the focus of inflammation. At the same time, their shear-thinning properties contribute to a qualitative distribution of the hydrogels on the skin surface or spreading out within injured joints for patients with osteoarthritis to improve joint lubrication and shock absorption.

The study of deformation properties of the synthesized hydrogels under compression and tension demonstrated that the HA–silica hydrogels had improved mechanical properties in comparison with HA (Table 3). Obviously, the main contribution to increasing the elasticity of the hybrid hydrogels under compression was made by a highly porous structure of the inorganic hydrogels. The SEM investigation of the synthesized hydrogels showed that, with an increase in HA molecular weight, the hybrid materials had a denser, less porous surface morphology (Figure 3). Therefore, the increasing molecular weight of HA and its loading led to a decrease in compressive deformation characteristics. The tensile Young’s moduli for the HA–silica hydrogels increased in comparison with both the inorganic hydrogels and HA. Most likely, this is due to HA–silica interactions.

On the whole, the resistance of the synthesized HA silica hydrogels to shear, compression, and tension is lower in comparison with the covalently cross-linking HA using organic agents [78,79,80] or alkoxysilanes [34,35,36,37]. However, topical or injectable delivery of anti-inflammatory drugs does not require hydrogels with very high mechanical properties. In addition, organic cross-linking agents are often toxic [81,82].

The duration of biological activity of HA including anti-inflammatory effects depends on the rate of its degradation in biological media. It is known that one of the ways to protect biomolecules from enzymatic degradation is their encapsulation in silica materials [83,84]. The results obtained in this work showed that the high molecular weight HA in the HA–silica hydrogels degrades more slowly than pure high molecular weight HA (Figure 8), i.e., the silica matrix is able to increase the resistance of the high molecular weight HA in the hybrid hydrogels to enzymatic degradation. The opposite effect was found for the low molecular weight HA.

The efficiency of HA as anti-inflammatory drug to a large extent depends on its retention time in the focus of inflammation. Due to the short period of functioning of HA in the body, it is necessary to inject it repeatedly in a short time to ensure a therapeutic effect. Therefore, it was important that the synthesized hydrogels released the acid over a long period of time according to the known kinetic law (preferably, according to zero-order kinetics).

In this work, a systematic study of the kinetics of HA release from the synthesized hydrogels was carried out. The results of the HA release study showed that the synthesized hydrogels were not able to attain the zero-order release of the drug. The release process followed the first order kinetics and was controlled by pseudo-Fickian diffusion. Delayed diffusion can occur due to the large size of HA molecules, their entanglement, as well as the interactions of HA with the silica matrix in the hybrid hydrogels. Such a mechanism promotes slowdown and prolongation of HA release in the studied media. The release rate can be controlled by HA loading and its molecular weight. Note that studies reporting the kinetic parameters and mechanisms of HA release from biomaterials or its permeation are rare. The diffusion mechanism of HA release was found in [85,86].

## 5. Conclusions

In summary, the studies have shown that silica hydrogels are a promising platform for development of soft formulations for topical and injectable delivery of hyaluronic acid, which is used for treatment of inflammatory disorders such as arthritis, skin diseases (atopic dermatitis, psoriasis, eczema, etc.), bowel diseases, etc.:the HA–silica hydrogels are based on low toxicity, biocompatible, biodegradable colloid silica, which is already used in pharmaceuticals and cosmetics;they exhibit pseudoplastic behavior and thixotropic properties, which determine the convenience and safety of their administration (injection or application of the hydrogel products to the surface of the human skin);the hybrid HA–silica hydrogels possess improved mechanical strength in comparison with pure HA;the HA–silica hydrogels containing high molecular weight HA are less susceptible to enzymatic degradation than pure HA;the drug release follows first order kinetics and is controlled by pseudo-Fickian diffusion;synthesis of the hydrogels is inexpensive.

## Figures and Tables

**Figure 1 pharmaceutics-15-00077-f001:**
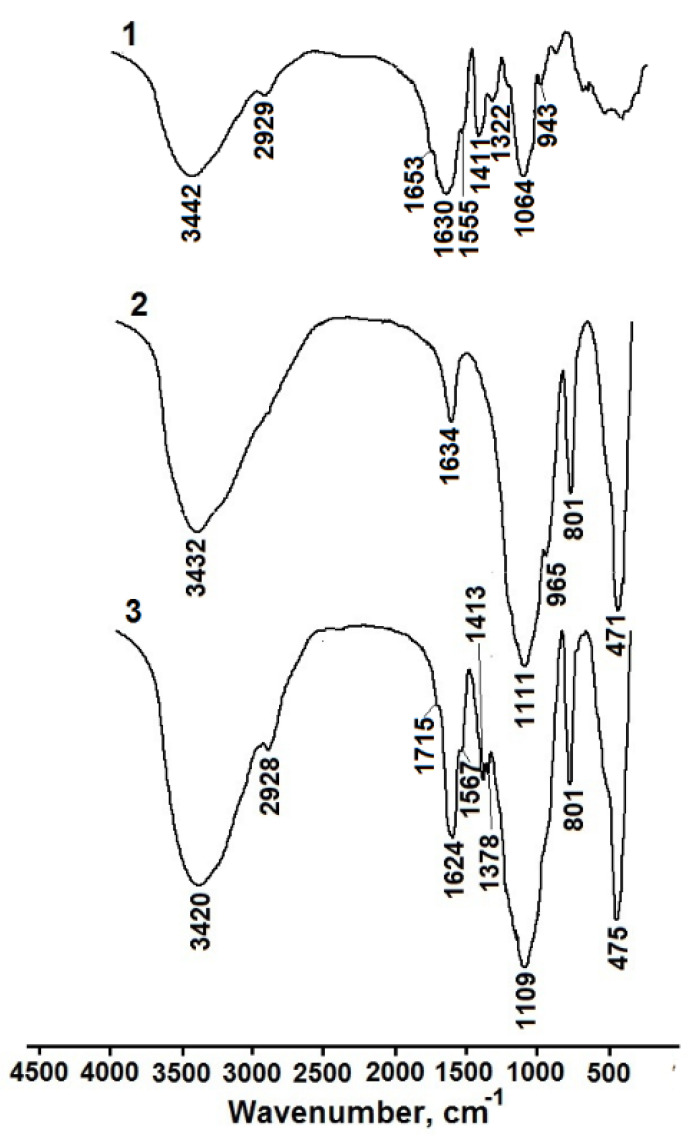
FTIR spectra of crystalline HA (1), dried HG1 (2), and dried HG1(2%)l (3).

**Figure 2 pharmaceutics-15-00077-f002:**
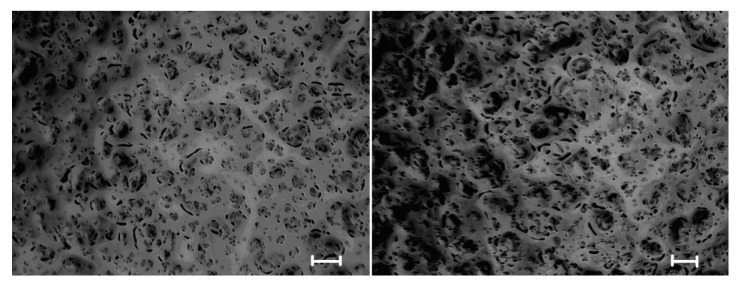
Optical microscopy images of HG2(1%)h (**left**) and HG2(1%)l (**right**) (Scale bars are 250 μm).

**Figure 3 pharmaceutics-15-00077-f003:**
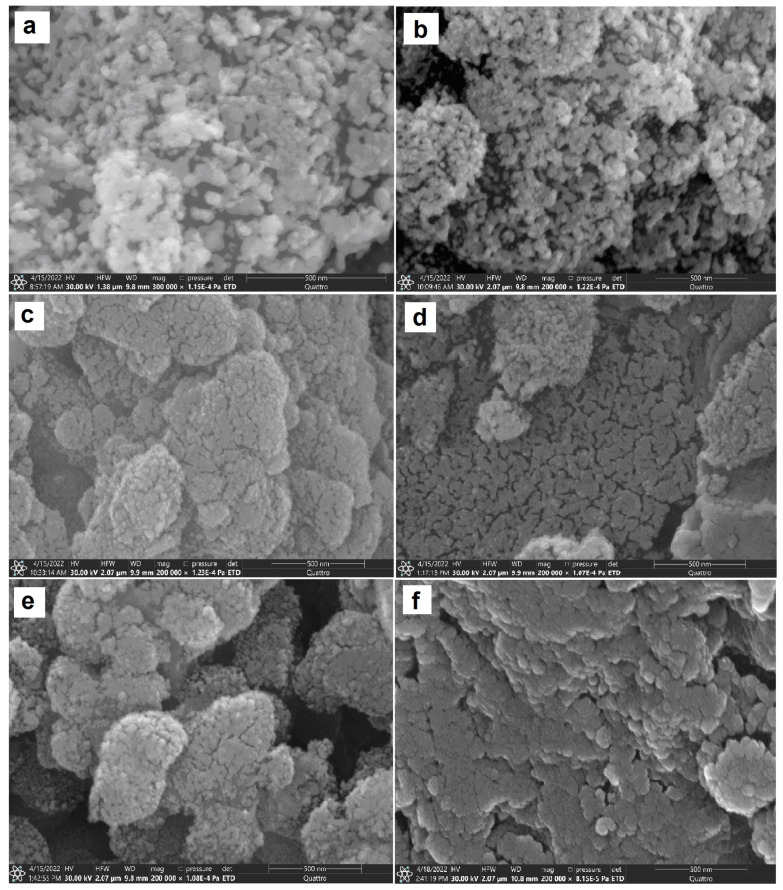
SEM images of HG1 (**a**), HG2(**b**), HG1(2%)l (**c**) HG2(2%)l (**d**), HG1(2%)h (**e**), and HG2(2%)h (**f**).

**Figure 4 pharmaceutics-15-00077-f004:**
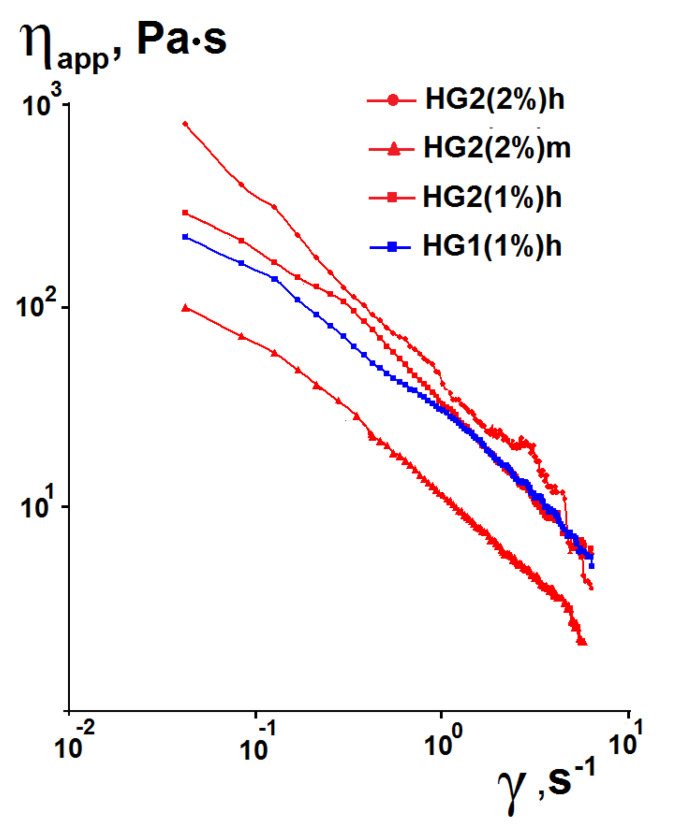
Dependencies of apparent viscosity on shear rate for some synthesized hydrogels.

**Figure 5 pharmaceutics-15-00077-f005:**
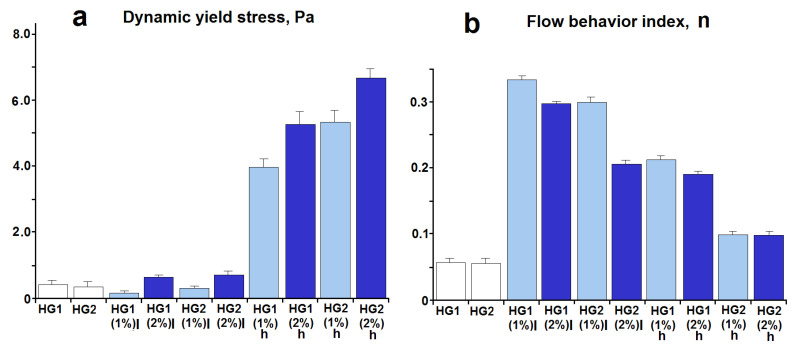
Comparative diagrams of dynamic yield stress and flow behavior index for synthesized hydrogels.

**Figure 6 pharmaceutics-15-00077-f006:**
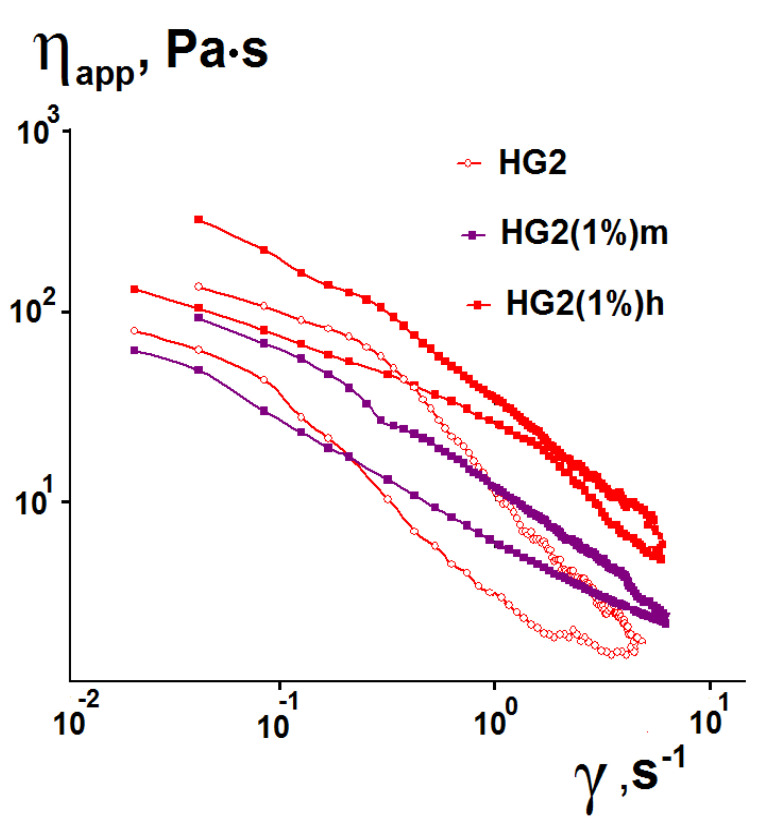
Hysteresis loops for some hydrogels.

**Figure 7 pharmaceutics-15-00077-f007:**
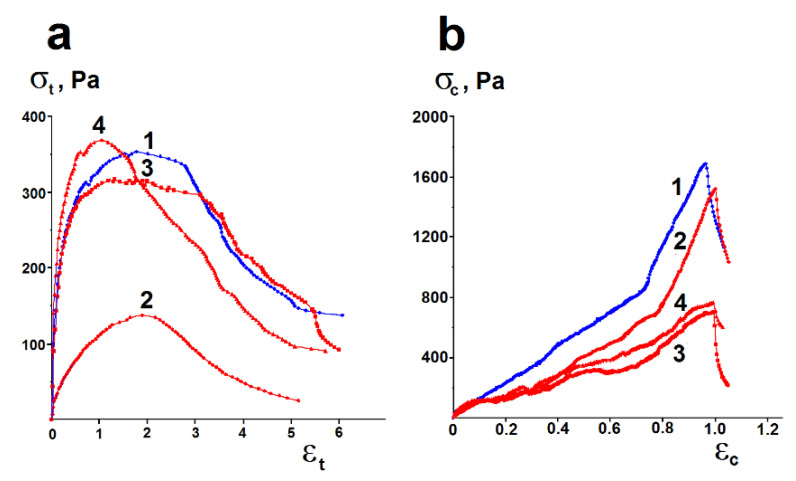
Experimental tensile (**a**) and compressive (**b**) stress–strain curves for HG1(2%)l (1), HG2(2%)l (2), HG2(2%)h (3), and HG2(1%)h (4).

**Figure 8 pharmaceutics-15-00077-f008:**
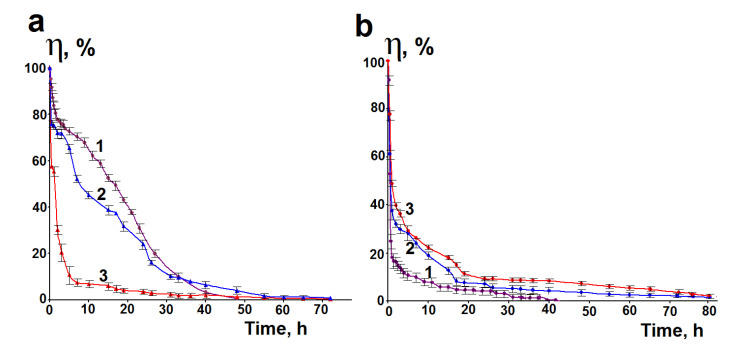
Degree of apparent viscosity loss of HA and HA–silica hybrid hydrogels as function of incubation time with hyaluronidase (37 °C; shear rate 0.5 s^−1^; enzyme concentration 22 U/mL of hydrogel). (**a**) 1-low molecular weight HA (2%), 2-HG1(2%)l, 3-HG2(2%)l and (**b**) 1-high molecular weight HA (2%), 2-HG1(2%)h, 3-HG2(2%)h.

**Figure 9 pharmaceutics-15-00077-f009:**
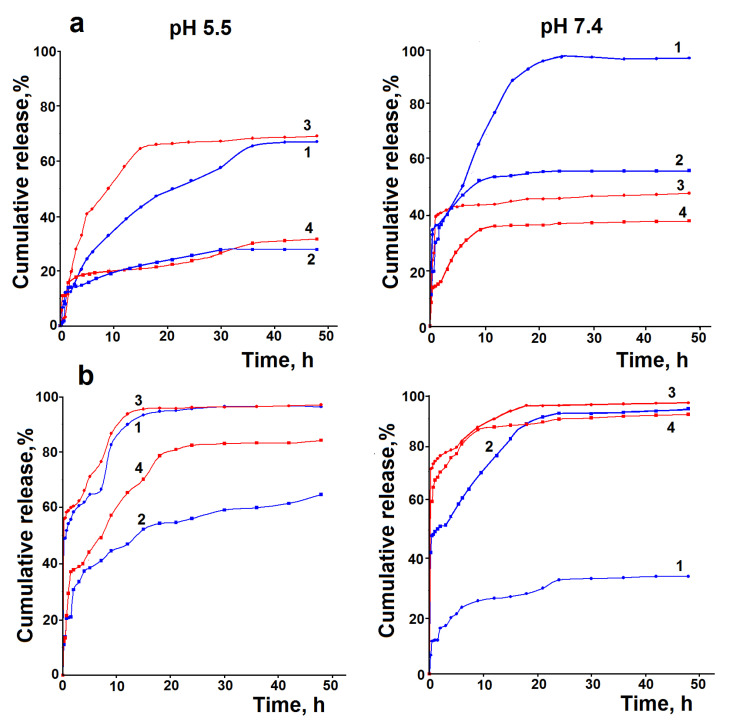
Experimental release profiles of HA from synthesized hydrogels in the media with pH 5.5 (32 °C) and 7.4 (37 °C): (**a**) 1-HG1(1%)l; 2-HG1(2%)l; 3-HG2(1%)l; and 4-HG2(2%)l; (**b**) 1-HG1(1%)h; 2-HG1(2%)h; 3-HG2(1%)h; and 4-HG2(2%)h. (The data are presented as an average of three measurements.)

**Table 1 pharmaceutics-15-00077-t001:** Designation and synthesis conditions of hydrogel materials (l- low molecular weight HA, h- high molecular weight HA).

Hydrogel	HCl Concentration	HA MW	Sample of HA, g	HA Loading, mg/g of Hybrid Hydrogel	Final pH
HG1	0.030 M	-	-	-	6.47
HG2	0.125 M	-	-	-	6.35
HG1(1%)l	0.030 M	50–100 kDa	0.5131	9.27	6.58
HG2(1%)l	0.125 M	50–100 kDa	0.5068	9.30	6.39
HG1(2%)l	0.030 M	50–100 kDa	1.0196	18.45	6.46
HG2(2%)l	0.125 M	50–100 kDa	1.0177	18.72	6.37
HG1(1%)h	0.030 M	1.0–1.5 MDa	0.5189	9.65	6.53
HG2(1%)h	0.125 M	1.0–1.5 MDa	0.5129	9.53	6.44
HG1(2%)h	0.030 M	1.0–1.5 MDa	1.0127	18.70	6.76
HG2(2%)h	0.125 M	1.0–1.5 MDa	1.0367	19.25	6.52

**Table 2 pharmaceutics-15-00077-t002:** Thixotropic index of synthesized hydrogels.

Hyd-Rogel	HG1	HG2	HG1 (1%)l	HG1 (2%)l	HG2 (1%)l	HG2 (2%)l	HG1 (1%)h	HG1 (2%)h	HG2 (1%)h	HG2 (2%)h
T	0.469	0.518	0.300	0.396	0.399	0.408	0.332	0.394	0.360	0.549

**Table 3 pharmaceutics-15-00077-t003:** Deformation properties of hydrogels under compression and tension (the data are presented as an average ± SD, n = 5).

Hydrogel	Tensile YM, kPa	Ultimate Tensile Strength, kPa	Compressive YM, kPa	Ultimate Compressive Strength, kPa
HA (1%) l ^1^	0.11 ± 0.03	0.065 ± 0.007	0.10 ± 0.02	0.11 ± 0.02
HA(1%) h ^1^	0.21 ± 0.04	0.010 ± 0.003	0.15 ± 0.02	0.12 ± 0.02
HG1	0.14 ± 0.03	0.11 ± 0.03	10.72 ± 0.88	38.64 ± 3.1
HG2	0.24 ± 0.04	0.06 ± 0.01	8.73 ± 0.61	36.54 ± 3.5
HG1(1%) l	1.46 ± 0.08	0.28 ± 0.02	2.09 ± 0.06	12.8 ± 1.5
HG1(2%) l	1.35 ± 0.16	0.37 ± 0.03	1.01 ± 0.03	1.78 ± 0.12
HG2(1%) l	1.23 ± 0.11	0.29 ± 0.02	5.59 ± 0.33	24.20 ± 2.2
HG2(2%) l	0.31 ± 0.08	0.14 ± 0.01	0.84 ± 0.03	1.64 ± 0.12
HG1(1%) h	1.52 ± 0.11	0.32 ± 0.02	0.26 ± 0.04	0.64 ± 0.09
HG1(2%) h	0.82 ± 0.06	0.14 ± 0.01	0.24 ± 0.03	0.36 ± 0.03
HG2(1%) h	1.81 ± 0.17	0.37 ± 0.02	0.52 ± 0.06	0.91 ± 0.08
HG2(2%) h	1.37 ± 0.08	0.31 ± 0.01	0.46 ± 0.03	0.93 ± 0.07

^1^ HA l and HA h are the low molecular weight and the high molecular weight hyaluronic acid.

**Table 4 pharmaceutics-15-00077-t004:** Kinetic parameters of HA release from synthesized hydrogels.

Hydrogels	pH 5.5 (32 °C)			pH 7.4 (37 °C)		
	Burst Effect, %	First Order Model	Korsmeyer–Peppas Model	Burst Effect, %	First Order Model	Korsmeyer–Peppas Model
HG1(1%) l	12.5	k_1_ = 7.1 × 10^−3^ h^−1^R^2^ = 0.9785	n = 0.40 k_K–P_ = 3.07 h^−n^ R^2^ = 0.9603	39.9	k_1_ = 6.1·10^−3^ h^−1^ R^2^ = 0.9833	n = 0.28 k_K–P_ = 2.18 h^−n^ R^2^ = 0.9622
HG1(2%) l	1.8	k_1_ = 5.1 × 10^−3^ h^−1^R^2^ = 0.9971	n = 0.22 k_K–P_ = 1.57 h^−n^ R^2^ = 0.9603	40.2	k_1_ = 7.7 × 10^−3^ h^−1^ R^2^ = 0.9912	n = 0.29 k_K–P_ = 2.22 h^−n^ R^2^ = 0.9722
HG2(1%) l	3.2	k_1_ = 7.4 × 10^−3^ h^−1^R^2^ = 0.9815	n = 0.42 k_K–P_ = 3.15 h^−n^ R^2^ = 0.9603	39.3	k_1_ = 6.5 × 10^−3^ h^−1^ R^2^ = 0.9988	n = 0.38 k_K–P_ = 2.87 h^−n^ R^2^ = 0.9745
HG2(2%) l	11.1	k_1_ = 5.4 × 10^−3^ h^−1^R^2^ = 0.9969	n = 0.29 k_K–P_ = 2.03 h^−n^ R^2^ = 0.9603	16.5	k_1_ = 5.8 × 10^−3^ h^−1^ R^2^ = 0.9979	n = 0.25 k_K–P_ = 1.87 h^−n^ R^2^ = 0.9695
HG1(1%) h	49.4	k_1_ = 8.5 × 10^−3^ h^−1^R^2^ = 0.9806	n = 0.45 k_K–P_ = 3.97 h^−n^ R^2^ = 0.9603	12.1	k_1_ = 5.6 × 10^−3^ h^−1^ R^2^ = 0.9968	n = 0.24 k_K–P_ = 1.66 h^−n^ R^2^ = 0.9718
HG1(2%) h	20.9	k_1_ = 7.4 × 10^−3^ h^−1^R^2^ = 0.9859	n = 0.35 k_K–P_ = 2.33 h^−n^ R^2^ = 0.9603	50.6	k_1_ = 7.6 × 10^−3^ h^−1^ R^2^ = 0.9956	n = 0.40 k_K–P_ = 2.94 h^−n^ R^2^ = 0.9725
HG2(1%) h	56.8	k_1_ = 8.3 × 10^−3^ h^−1^R^2^ = 0.9871	n = 0.43 k_K–P_ = 3.34 h^−n^ R^2^ = 0.9603	63.0	k_1_ = 7.1 × 10^−3^ h^−1^ R^2^ = 0.9865	n = 0.36 k_K–P_ = 2.45 h^−n^ R^2^ = 0.9625
HG2(2%) h	37.4	k_1_ = 7.8 × 10^−3^ h^−1^R^2^ = 0.9918	n = 0.39 k_K–P_ = 3.0 h^−n^ R^2^ = 0.9603	75.2	k_1_ = 5.6 × 10^−3^ h^−1^ R^2^ = 0.9955	n = 0.27 k_K–P_ = 1.92 h^−n^ R^2^ = 0.9803

## Data Availability

Not applicable.

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
