# Peer review of "Silica Hydrogels as Platform for Delivery of Hyaluronic Acid"

_pharmaceutics, 2022, doi:10.3390/pharmaceutics15010077_

Round 1

Reviewer 1 Report

Parfenyuk and Dolinina in the article describe silica hydrogels as a platform for delivery of hyaluronic acid. The article is interesting and introduces an elements of novelty. Hyaluronic acid is becoming more and more popular in cosmetology and research on it is very desirable.

I suggest making some minor corrections:

1) keywords should be in alphabetical order,

2) the introduction is written nicely, but I suggest to include a few sentences about the penetration of HA through the skin. Are there any reports in this regard?

3) in the abstract on line 8 there are too many dots.

4) at what temperature the hydrogels were stored.

In the future, the authors might consider testing in vitro HA permeation through the human or pig skin using a Franz diffusion chamber.

I recommend for publication.

Author Response

Comment. Keywords should be in alphabetical order. Response. The correction made to the text.

Comment. The introduction is written nicely, but I suggest to include a few sentences about the penetration of HA through the skin. Are there any reports in this regard?Response. The information about penetration of HA through the skin added to the text. Please, see page 2. Marked in green.

Comment. In the abstract on line 8 there are too many dots Response. The indicated error is corrected.

Comment. At what temperature the hydrogels were stored.Response. The hydrogels were stored at room temperature.

Comment. In the future, the authors might consider testing in vitro HA permeation through the human or pig skin using a Franz diffusion chamber.Response. The measurements will be completed soon . 

Reviewer 2 Report

The present paper deals with the formation of silica hydrogels in order to be used as a delivery system for hyaluronic acid (HA). The developed delivery system is characterized with respect to morphology, rheological and deformation properties as well as regarding the release kinetics of HA from the synthesized hybrid hydrogels and its enzymatic degradation. The current study could be of interest to the scientific society since it presents valuable information on the characteristics of HA-silica hydrogels to be used as a platform for the delivery of HA. On the other hand, the overall quality of the manuscript is substandard (e.g., use of English language, quality of writing, coherence, clarity of the scientific presentation). The numerous syntax errors make reading and understanding difficult and, in some points, completely unclear in terms of what is meant. Accordingly, the authors need to revise their manuscript in this respect and to take into account the following comments in order to be considered for publication:

· The authors should clarify whether, for the formation of pure silica hydrogels, the prepared sol was added dropwise to 50 mL of a phosphate buffer solution (pH 7.4).

· The authors should define the abbreviations (HG1, HG2, HG1(1%)l, HG2(%)l, HG1(1%)h, etc.) presented for the first time in Table 1.

·  The authors should indicate whether they used a mold in order to form hydrogels with specific dimensions and shape.

·  Figure 4 should be presented in logarithmic scale (Log-Log) so that the readers can better interpret the presented results.

·  Similarly, Figure 6 should be constructed in logarithmic scale.

·   In the manuscript it is reported that the best fit for both silica and hybrid hydrogels was observed when using the Herschel-Bulkley model. The authors should indicate whether this is the first time that silica and silica-HA hydrogels are best fitted with this model.  

·  The authors should extensively discuss the results presented in Figure 7 and Table 3.

·  In order to better interpret the results presented in Figure 8 (a,b), the authors should provide the apparent viscosity curves for some indicative time points (e.g., 5, 10, 20, 30 and 40h) for the following: (a) 1-low molecular weight HA (2%), 2-HG1(2%)l, 3-HG2(2%)l (b) 1-high molecular weight HA (2%), 2-HG1(2%)h, 3-HG2(2%)h.

·    Figure 9: The authors should clearly indicate the release data they fitted to the Korsmeyer-Peppas model. It should be noted that the Korsmeyer-Peppas model can be strictly applied to experimental data corresponding to <60% of drug release [Ritger and Peppas, Journal of Controlled Release, 5 (1987) 37-42]

·    Figure 9: The legend should be corrected, since (b) refers to high molecular weight.

·     Lines 10-12: “The main disadvantages that limit its use are its exposure to rapid biodegradation and low mechanical properties.” should be changed to “The main disadvantages that limit its use are its low mechanical properties and its rapid biodegradation.”

·     Line 160: The phrase “The solution was kept for a day to swell the acid.” should be removed.

·      Line 382: “yield strength (τ0)” should be changed to “yield stress (τ0)”

Author Response

Comment.  The authors should clarify whether, for the formation of pure silica hydrogels, the prepared sol was added dropwise to 50 mL of a phosphate buffer solution (pH 7.4).

Response. Thank you. This synthesis detail is added to the text. Please, see page 4, section 2.2 (Marked in green)

Comment. The authors should define the abbreviations (HG1, HG2, HG1(1%)l, HG2(%)l, HG1(1%)h, etc.) presented for the first time in Table 1.

Response. Table 1 is given for deciphering of the indicated abbreviations. As can be seen from the table, the designation “1” and “2” refer to the hydrogels prepared using different concentration of HCl (1-0.03M  and 2-0.125M). The designations “1%” and “2%” refer to HA loading. The deciphering of “l” and “h” is given in the caption to the Table.

Comment. The authors should indicate whether they used a mold in order to form hydrogels with specific dimensions and shape

Response. We didn’t use a mold to form hydrogels with specific dimensions and shape. The  hydrogels.  The formation of the hydrogels occurred in glass flasks of the required volume.

Comment. Figure 4 should be presented in logarithmic scale (Log-Log) so that the readers can better interpret the presented results.  Similarly, Figure 6 should be constructed in logarithmic scale

Response. We have presented Figure 4 and Figure 6 in logarithmic coordinates

Comment. In the manuscript it is reported that the best fit for both silica and hybrid hydrogels was observed when using the Herschel-Bulkley model. The authors should indicate whether this is the first time that silica and silica-HA hydrogels are best fitted with this model

Response. First of all, it should be noted that the investigations of properties of silica-based hydrogels as individual materials are rare. There are many works in literature devoted to study kinetics of gelation process through rheological properties (for example, ref [33], [34], [37], E. Katoueizadeh et al. J. Mater. Res. Technol. 2020, 9, 10157–10165 DOI:

10.1016/j.jmrt.2020.07.020).The effects of catalyst, salt additives, crosslinkers, etc. on gelation time are most often studied. Rheological properties of silica and HA-silica hydrogels were studied using dynamic rheology oscillatory) method (for example, B.A. Serban et al. Gels 2020, 6, 38 DOI:10.3390/gels6040038, ref [35], [36], [37]). The steady state rheological techniques and mathematical flow models, including Herschel-Bulkley model, were often used for description of rheological behavior of various polymer composite hydrogels (for example, Z. Fun et al. Int. J. Biol. Macromol. 2022, 208, 1-10 DOI: /10.1016/j.ijbiomac.2022.03.054; H. Herrada-Manchón et al.  Gels 2022, 8, 28. https://doi.org/10.3390/gels8010028; M. V. Ghica et al. Molecules 2016, 21, 786; doi:10.3390/molecules21060786).

In the literature we found study on the application of the model to describe the rheological properties of inorganic hydrogels (F. García-Villén et al. Applied Clay Science 197 (2020) 105772 DOI: 10.1016/j.clay.2020.105772). Therefore, it may be concluded that the model was applied for the first time.

Comment.· The authors should extensively discuss the results presented in Figure 7 and Table 3.

Response. We have inserted some additional explanations in the text. Please, see page 12-13, section 3.5. (Marked in green).

Comment.  In order to better interpret the results presented in Figure 8 (a,b), the authors should provide the apparent viscosity curves for some indicative time points (e.g., 5, 10, 20, 30 and 40h) for the following: (a) 1-low molecular weight HA (2%), 2-HG1(2%)l, 3-HG2(2%)l (b) 1-high molecular weight HA (2%), 2-HG1(2%)h, 3-HG2(2%)h.

Response. To take into account the contribution of the shear-thinning effect to the loss of apparent viscosity, the curves were obtained for the samples without the enzyme addition under the same experimental conditions. Due to the low shear rate, the thixotropy of the samples, as well as the absence of shear load between measurements at a certain time (the measurements were carried out for 8 min), the viscosity loss in percent was very low (1-2%). If we present the curves in the figure, they will coincide. So we decided to give this information in the text. Please, see  page 6, section 2.3.5 (Marked in green) 

Comment.  Figure 9: The authors should clearly indicate the release data they fitted to the Korsmeyer-Peppas model. It should be noted that the Korsmeyer-Peppas model can be strictly applied to experimental data corresponding to <60% of drug release [Ritger and Peppas, Journal of Controlled Release, 5 (1987) 37-42]

Response. This is indicated on the page 6, where the used models for description of the experimental release profiles are shown (eq. 12)

Comment.    Figure 9: The legend should be corrected, since (b) refers to high molecular weight.

Response. We apologize. This technical error is corrected

Comment. Line 10-12. “The main disadvantages that limit its use are its exposure to rapid biodegradation and low mechanical properties.” should be changed to “The main disadvantages that limit its use are its low mechanical properties and its rapid biodegradation.” 

Response. This is corrected. (Marked in green)

Comment.    Line 160: The phrase “The solution was kept for a day to swell the acid.” should be removed.

Response. We do not agree with this comment. It is important for us that the properties of the polymer do not change during their preparation, aging and storage.

Comment.  ·      Line 382: “yield strength (τ0)” should be changed to “yield stress (τ0)”

Response.  This misprint is corrected.

Reviewer 3 Report

This manuscript reported that the hybrid HA-silica hydrogels were synthesized by sol-gel method. Morphology, deformation properties, their resistance to enzymatic degradation in vitro, and kinetics and mechanisms of HA release from the hybrid hydrogels were investigated. The design idea of this manuscript is novel and the experimental data are sufficient. However, unreasonable language expression, too many detailed errors, and unattractive drawing technique and layout of figures and tables severely affect the quality of this manuscript. Therefore, the manuscript needs minor revisions before possible publication in Pharmaceutics. The comments are provided below:

1.      Some grammatical mistakes should be carefully checked, such as lines 7 - 8, “Hyaluronic acid (HA) is chondroprotective and anti-inflammatory drug used clinically fir treatment of inflammatory disorders (arthritis, skin diseases, bowel diseases, etc.). .).” should be changed as “Hyaluronic acid (HA) is a chondroprotective and anti-inflammatory drug used clinically for treatment of inflammatory disorders (arthritis, skin diseases, bowel diseases, etc.).”. same problems are found line 133.

2.      For the “Introduction” section, it is so cumbersome. The author should be rewritten for easy reading. Please carefully check the grammar of the sentences and the spelling of the words.

3.      In the Table 1 to 3, there are garbled characters between Tables and line numbers, the author is advised to adjust these Tables format. In addition, please retain the same significant digits for HCl concentration in the Table 1.

4.      In the sections of “3. Results and Discussion” and “4. Discussion”, there are some overlaps in the title and content, the author is advised to make corresponding modifications.

5.      All images and captions should be in the same format. For example, Figures 5 should be bold. Furthermore, each image should have a corresponding serial number (a, b, c...), and label it at the top left corner.

6.      Authors are advised to examine every part of the manuscript carefully, the representation of all images in the manuscript should be match the files provided, such as “Figure 1” instead of “Fig. 1”.

7.      Suggested edits to the text:

a.      Page 2, line 47, change “skin repairmen” to “skin repairment”.

b.      Page 2, line 78, change “crossinked” to “crosslinked”.

c.      Page 3, line 120, change “manufacture” to “manufacturing”.

d.      Page 3, line 135, change “320C and 370C” to “32 °C and 37 °C”, same problems are found in whole manuscript, and check them carefully please.

e.      Page 4, line 188, change “cm-1” to “cm-1(superscript)”.

f.       Page 10, line 384, change “a” to “the”.

g.      In Table 1, change “HCL” to “HCl”.

Author Response

Comment. Some grammatical mistakes should be carefully checked, such as lines 7 - 8, “Hyaluronic acid (HA) is chondroprotective and anti-inflammatory drug used clinically fir treatment of inflammatory disorders (arthritis, skin diseases, bowel diseases, etc.). .).” should be changed as “Hyaluronic acid (HA) is a chondroprotective and anti-inflammatory drug used clinically for treatment of inflammatory disorders (arthritis, skin diseases, bowel diseases, etc.).”. same problems are found line 133.

Response. The misprint and Ehglish style are corrected. (marked in green)

Comment. For the “Introduction” section, it is so cumbersome. The author should be rewritten for easy reading. Please carefully check the grammar of the sentences and the spelling of the words.

Response. We have two opposite opinions of the reviewers. Therefore, the Introduction is left unchanged.

Comment. In the Table 1 to 3, there are garbled characters between Tables and line numbers, the author is advised to adjust these Tables format. In addition, please retain the same significant digits for HCl concentration in the Table 1.

Response. We cannot correct this. Line numbering appears automatically after loading of our template.

Comment. In the sections of “3. Results and Discussion” and “4. Discussion”, there are some overlaps in the title and content, the author is advised to make corresponding modifications

Response. Thank you. The error is corrected

Comment.      All images and captions should be in the same format. For example, Figures 5 should be bold. Furthermore, each image should have a corresponding serial number (a, b, c...), and label it at the top left corner

Response. The figures and caption to Figure 5 are corrected

Comment. Suggested edits to the text:

  1. Page 2, line 47, change “skin repairmen” to “skin repairment”.
  2. Page 2, line 78, change “crossinked” to “crosslinked”.
  3. Page 3, line 120, change “manufacture” to “manufacturing”.
  4. Page 3, line 135, change “320C and 370C” to “32 °C and 37 °C”, same problems are found in whole manuscript, and check them carefully please.
  5. Page 4, line 188, change “cm-1” to “cm-1(superscript)”.
  6. Page 10, line 384, change “a” to “the”.
  7. In Table 1, change “HCL” to “HCl”.

Response. The indicated errors are corrected.